# RFID Application in a Multi-Agent Cyber Physical Manufacturing System

**Maryam Farsi [1,*]**, **Christina Latsou [1]**, **John Ahmet Erkoyuncu [1]** and **Geoffrey Morris [2]**

1    School of Aerospace, Transport and Manufacturing, Cranfield University, Cranfield MK43 0AL, UK; christina.latsou@cranfield.ac.uk (C.L.); j.a.erkoyuncu@cranfield.ac.uk (J.A.E.)
2    Cryogatt Systems Limited, Kings Langley WD4 8FR, UK; geoff.morris@cryogatt.com
*    Correspondence: maryam.farsi@cranfield.ac.uk; Tel.: +44-123-475-0111

**Abstract:** In manufacturing supply chains with labour-intensive operations and processes, individuals perform various types of manual tasks and quality checks. These operations and processes embrace engagement with various forms of paperwork, regulation obligations and external agreements between multiple stakeholders. Such manual activities can increase human error and near misses, which may ultimately lead to a lack of productivity and performance. In this paper, a multi-agent cyber-physical system (CPS) architecture with radio frequency identification (RFID) technology is presented to assist inter-layer interactions between different manufacturing phases on the shop floor and external interactions with other stakeholders within a supply chain. A dynamic simulation model in the AnyLogic software is developed to implement the CPS-RFID solution by using the agent-based technique. A case study from cryogenic warehousing in cell and gene therapy has been chosen to test the validity of the presented CPS-RFID architecture. The analyses of the simulation results show improvement in efficiency and productivity, in terms of resource time-in-system.

**Keywords:** cyber physical system; agent-based simulation; RFID; cryogenic warehousing; complex systems; supply chain; cell and gene therapy

---

## 1. Introduction

Over the last decade, the deployment of radio frequency identification (RFID) within different supply chains has had a major influence on the traceability of materials and information. In the manufacturing sector, RFID technology improves the trackability of production processes, and the integrity of information flows on a shop floor. In warehousing, RFID enhances inventory management and streamlines its interactions with other logistical components, through the improved track and trace of materials and transporters. Moreover, RFID technology, together with connected IT platforms, has a crucial impact on control systems within businesses. The application of such digital technologies can ultimately lead to significant performance and productivity improvements for businesses [1,2].

RFID readers use radio waves to capture information and can be categorised into two types: passive and active. Passive RFID tags do not have an internal power source, and they are equipped with an electromagnetic chip that is read by a battery-powered RFID reader. These tags are relatively small and have a long lifespan. However, such tags can be read at only short distances (i.e., a few feet), which limits their use for certain applications. Active RFID tags have their own source of power. These tags can be read at more than a hundred feet distance. However, such tags have a limited lifespan due to battery constraints and are more expensive. Regardless of the RFID type, implanting such technologies advances the level of maturity in digitalisation, automation and real-time data capture. Moreover, RFID technology increases information visibility, contributes to lean manufacturing, safety and security.

Additionally, the technology can potentially reduce manual handling and labour-intensive activities, operation downtime, and human errors.

The cyber physical manufacturing system (CPMS) and cyber physical production system (CPPS) can be described as an integrated IT platform for interactive computation, networking and physical processes on a shop floor or a supply chain. The Internet of Things (IoT) provides the infrastructure for CPS implementation. CPS uses several devices such as sensors, RFID readers, and mobile phones to provide interactivity in different applications [3]. In the manufacturing industry, CPS can support businesses to retain their competitiveness by enabling product development optimisation, production system control, real-time informed decisions, and deployment of manufacturing smart systems [4,5]. In addition, within a supply chain, manufacturers are required to effectively collaborate with various stakeholders to be able to mitigate risks emerging from market diversity and customer requirements. In this regard, CPS can enable social manufacturing to allow mass customisation and to improve logistics transparency based on 'customised community space configuration' [6].

Acquiring essential data and maintaining information integrity within manufacturing processes and supply chains are challenging and time-consuming tasks, due to the amount of paperwork and manual audit procedures involved. In many small and medium enterprises (SMEs), the application of automated techniques such as IoT, CPS and RFID is not mature and is at the early stages of proof-of-concept. Although such advancements create new challenges for industries in terms of uncertainty management [7], security [8], cloud computing [9] and big data management [10], evaluating the impact of such technologies on supply chains' and businesses' performance is crucial for any investment appraisal.

For a system of systems, such as supply chains and complex manufacturing systems, the Agent-Based Modelling (ABM) technique is a promising approach for the system design and system simulation [11]. To explore and substantiate the RFID application in CPMS, this paper addresses the following research question: "How can the ABM technique be applied to develop an RFID-CPMS architecture?". Moreover, this study contributes to knowledge by extending the application of ABM technique to develop a multi-agent architecture for an RFID-CPS. The context of this research is the application of RFID technology in a CPMS. We have also constructed a number of Unified Modelling Language (UML) diagrams to depict different perspectives and layers of the multi-agent RFID-CPS in a cryogenic supply chain.

This paper is structured as follows: Section 2 provides the theory and background of RFID and CPS architecture in manufacturing and supply chains. The research methodology to carry out this study is presented in Section 3. The multi-agent CPS architecture with the application of RFID is developed in Section 4. This is followed by a case-study in a cryogenic supply chain in this section. Section 5 provides the discussion on the multi-agent RFID-CPS architecture. The concluding remarks and future work are presented in Section 6.

## 2. Theory

With the rapid development of digital and information technologies within manufacturing sectors and supply chains, the novel application of IoT, CPS and RFID data management has been the focus of many studies in the past two decades. In manufacturing, CPPS is an emerging information system that, together with IoT and cloud computing technologies, supports automation and digital maturity [4,12]. In this regard, Ding et al. [6] presented an RFID-enabled social manufacturing system platform for real-time data monitoring, dispatching and transportation. They argued that such a platform could be deployed for data sharing and collaborative decision-making between different stakeholders.

IoT infrastructures and CPS frameworks for the application of RFID sensors for security and traceability purposes have been studied previously in the context of smart cities [13,14], logistics [15], small manufacturing [16], production [17], smart manufacturing [5,18], automotive industry [19], and product-service systems [20]. Suhail et al. [21] discussed the data provenance and security challenges associated with an RFID-based IoT. The data provenance challenges are defined as data

storage, processing, biding, interoperability and fault tolerance, whereas, the security challenges relate to data integrity, confidentiality, privacy, accessibility, freshness, availability, and establishment in an IoT network [21]. The reliability of the RFID tags is critical for information integrity. Untrustworthy tags can raise several IT and security challenges for the business. In this regard, Zhang et al. [22] proposed Bloom-Filter-Based Unknown Tag Identification (BUTI) to identify unknown RFID tags within a manufacturing system.

Cloud-based IoT network systems enhance the interaction and connectivity of CPS elements. A multi-layer cloud system, composed of an operating system, middleware, communication technologies and applications, requires a system engineering architecture with interactive and autonomous features [23]. In such systems, appropriate database design techniques can provide the opportunity to develop a multi-layer data architecture [24]. For instance, Hentout et al. [25] presented a multi-agent architecture for RFID-CPS in a multi-robotic system. They studied multiple scenarios with a different number of robots and RFID tags within a simulated environment to examine the robots' detection time. Moreover, Wang et al. [26] proposed a cloud system architecture to assist inter-layer interaction and inter-robot negotiation of a smart factory with RFID tagged products. In addition, an RFID-enabled positioning approach for smart factories was presented by Lu et al. [27]. They conducted a system simulation and implementation for evaluating positioning metrics in RFID-tagged systems on a shop floor.

Wang et al. [12] developed a hybrid RFID-enabled decentralised control system for a flexible smart workpiece manufacturing shop floor. They argued that the hybrid architecture improves the intelligence and flexibility of tasks on a shop floor. Murofushi and Tavares [28] demonstrated that RFID technology could be a basis for smart products in Industry 4.0, since RFID provides unique identification, data connection and data storage for products. In a more recent study, a product-driven control CPS architecture with RFID technology was proposed by Mihoubi et al. [29]. Their proposed system adopted a discrete-event simulation technique with multi-agent system implementation to construct the system. Furthermore, the application of RFID-CPS in automated guided vehicles (AGV) on manufacturing shop floors has been studied by many authors [26,27,30]. A passive RFID tag-based CPS approach for AGV was presented by Lu et al. [30] in order to evaluate the influences of tags in smart warehousing and manufacturing management. In their study, several factors such as antenna selection, RFID installation and sensor location were examined to design the RFID-CPS architecture. Moreover, Tran et al. [5] developed a smart CPMS and IoT platform using wireless sensory network (WSN) and RFID devices. They proposed that smart CPMS is capable of adapting to certain manufacturing changes with a response time of a few seconds. Recently, the digital twin (DT)-based CPS architecture has been a core focus of the literature in smart manufacturing and advanced information technology research area. In this regard, Liu et al. [31] proposed a systemic framework to provide guidelines for a rapid system configuration by integrating CPS and DT techniques. Several system design characteristics such as scalability, modularity, autonomy, and distributed cooperation were discussed in their study. Moreover, Park et al. [32] discussed the application of integrated DT-based CPS in personalised and customised production. They concluded that such an integrated approach is effective to enhance production performance in terms of advance planning, scheduling, control system and context awareness of processes.

Overall, within the context of manufacturing and production systems, the application of the multi-agent approach for CPS design has been limited to the development of an object-oriented architecture for individual products [12,25] or a set of products and machines on a shop floor [29]. Moreover, the concept of 'agent' has been only used to develop algorithms for individual products' functions with embedded RFID tags [26] and their control systems [5,33]. Complex manufacturing and supply chain systems, consist of several sub-systems (i.e., phases) and sub-sub-systems (i.e., components) that operate simultaneously and interact with each other over time [11]. This paper extends the application of agent-based approach and proposes a multi-agent CPS architecture for complex manufacturing and supply chain systems with embedded RFID technology.

## 3. Methodology

This paper implemented a systematic review methodology to study the existing literature on RFID and 'Cyber Physical System'. The Web of Science (WoS) and Scopus research repositories were searched with no lower time-limit and up to August 2020. The search process in WoS has been restricted to 'English Language' and to 'Cyber Physical*' and 'RFID' topics, with the following search string: "("Cyber Physical") AND TOPIC: ("RFID") Timespan: All years. Indexes: SCI-EXPANDED, SSCI, A&HCI, CPCI-S, CPCI-SSH, ESCI". The Scopus search strategy was also in accordance with the Scopus guidelines. The keywords used to perform the search activity were: 'Cyber Physical System' or 'CPS' or 'Cyber Physical*' or 'Cyber-physical Production System'. The following search string was therefore used to identify the relevant studies: "Cyber Physical" AND "Production System" AND "RFID" AND (LIMIT-TO (EXACTKEYWORD,"Cyber Physical System") OR LIMIT-TO (EXACTKEYWORD, "Cyber-physical Systems") OR LIMIT-TO (EXACTKEYWORD, "Cyber-physical Systems (CPS)") OR LIMIT-TO(EXACTKEYWORD, "Cyber Physical*") OR LIMIT-TO (EXACTKEYWORD, "Cyber-Physical System (CPS)") OR LIMIT-TO (EXACTKEYWORD, "Cyber Physical Systems") OR LIMIT-TO (EXACTKEYWORD, "Cyber-Physical Systems") OR LIMIT-TO (EXACTKEYWORD, "CPS") OR LIMIT-TO (EXACTKEYWORD, "CPPS") OR LIMIT-TO (EXACTKEYWORD, "Cyber-physical Production System")). The four-stage PRISMA 2009 flow diagram was then followed to identify and select articles. In the identification stage, the search yielded 212 documents in total. Afterwards, in the screening stage, articles duplicated in different databases were identified, and only a single copy was retained. Subsequently, in the screening stage, the titles and abstracts of the remaining documents were examined for their relevance. This resulted in the exclusion of 136 further studies. Next, the full-texts of articles were manually reviewed and assessed for their eligibility. At this stage, an additional 31 studies were excluded due to the lack of originality, novelty and being in another context, such as maintenance and health monitoring, service contract design, sustainability, risk assessment. Finally, a total of 45 articles were selected for review, as illustrated in Figure 1.

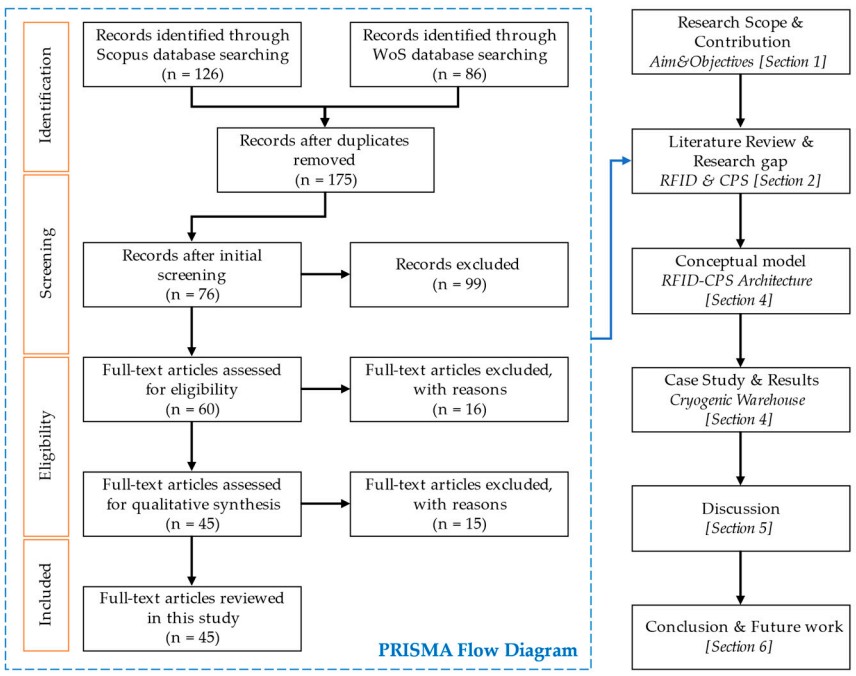

**Figure 1.** PRISMA flow diagram and research methodology.

The proposed multi-agent RFID-CPS architecture is presented based on the literature review and the theories behind RFID and CPS, as presented in Section 2. To validate the multi-agent architecture,

an industrial case study from a cryogenic warehouse company is considered. The agent-based simulation model of the case study with the embedded RFID was developed in the AnyLogic software, as presented in Section 4.

## 4. Results

A cold supply chain refers to the supply and logistics of perishable products and materials. In such a supply chain, evaluating the location and condition of transporters and shippers are vital. In addition, implementation of a logistics cyber-physical monitoring and control system optimisation supports shipment decisions and improve logistics efficiency [34]. In this study, a cryogenic supply chain in the cell and gene therapy (CGT) sector is studied to develop an RFID-CPS cyber architecture. In a cryogenic warehouse at the temperature of −190 °C, the advanced RFID tag and reader technologies [35–44] can be implemented for the recording, monitoring and auditing of cryo-materials on the shop floor. In such an environment, advanced RFID tags can considerably enhance inventory management, since the traditional labels can be easily frosted and become unreadable even after a short time period after their liquid nitrogen storage.

In this study, a cryogenic supply chain that is comprised of the manufacturer, distributor and the end-user is considered as a case study. The advanced RFID system is designed and installed at each stakeholder's site. The system architecture for RFID implementation is illustrated in Figure 2. According to this architecture, the system components are connected via the in-house network. A server PC, holding the system database, runs a web service that allows computers and mobile devices to be connected to the same network, to view and manipulate the recorded data. RFID readers located in the goods in and out areas are used to collect data on the arrival and dispatch of shippers, respectively. Similarly, the RFID readers situated in the cryogenic storage area are used to capture data for either storing cryo-materials after shippers' arrival or to retrieve cryo-materials preparing for dispatch.

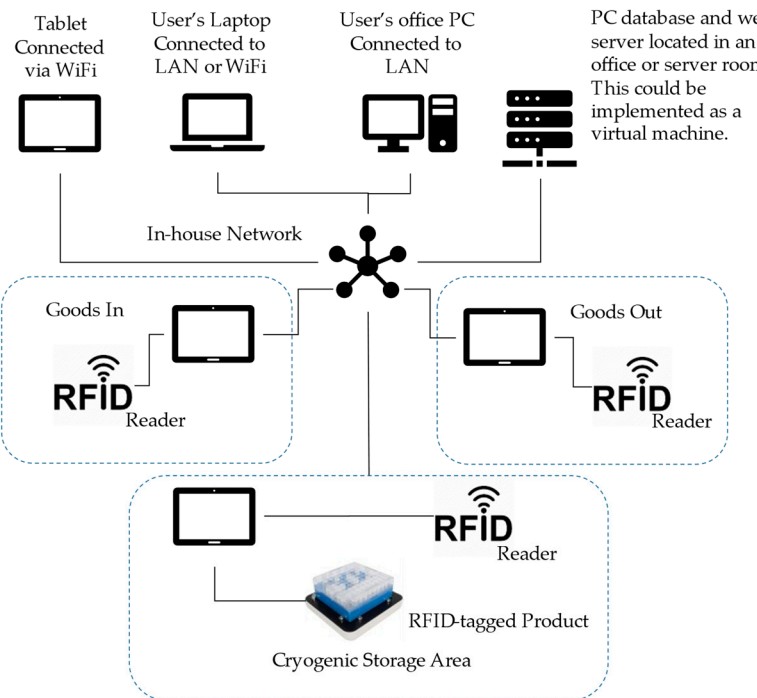

**Figure 2.** Radio frequency identification (RFID) system architecture.

### 4.1. Multi-Agent Cyber-Physical Manufacturing System

In this research work, a hybrid method that employs object—and process-oriented approaches using Agent-Based Modelling (ABM) and Discrete Event Simulation (DES) is proposed to model the

CGT enterprise architecture. Additionally, a built-in fully integrated database to read real-time data provided from RFID PC servers at each site within a cryogenic supply chain is deployed. The UML class diagram of the proposed multi-agent simulation model is presented in Figure 3. It should be noted that the CGT manufacturing system consists of multiple sub-systems that are able to operate simultaneously. In this work, these sub-systems are referred to as manufacturing phases. Each manufacturing phase entails various tasks that are described as goal-directed activities with a start and endpoint and an objective. Performing multiple tasks in different manufacturing phases can potentially lead to parallel interactions within the manufacturing system. A common example of parallel interactions is when operators with similar skills are required to perform a single task or a sequence of tasks in more than one manufacturing phase at the same time. The system architecture for the SCG supply chain consists of two main parts:

- Part I—ABM-DES hybrid system architecture: A system of multi-agent discrete events, inspired by [11], is developed to model a complex manufacturing system. Three levels of agents, macro, meso and micro, are used to capture the dynamic behaviour of the manufacturing system. The macro-level agent models the global manufacturing system, comprising manufacturing phases (meso-level agent) and manufacturing components (micro-level agent). Meso-level agents, modelled using the ABM approach, are employed to simulate the interactive structure of manufacturing phases. As seen from Figure 3, agents at this level are created as a single agent type always existing within the macro-agent environment. Micro-level agents, including human and equipment resources, data, information, etc., are also created using the ABM approach. Unlike meso-level agents, micro-agents are created as a population of agents of the same type living in the same environment. Following Figure 3, the behaviour of each agent at the micro-level belongs to a specific resource type (static, moving or portable), and may be characterised by specific capacity, rate and schedule. Function methods for describing algebraic rules and events for scheduling one-time or recurrent, concurrent or independent actions can also be defined. Finally, the DES modelling approach is employed for describing the discrete event states of each manufacturing phase modelled inside the meso-level agents.

- Part II—CPS-RFID system architecture: A database element, representing an actual database, is created to ensure communication with the agents described in Part I. The database element is associated with a database file (MS Access database/MS Excel spreadsheet) that holds data captured from the RFID system. An absolute file path is created, ensuring the connection between the database element and the database file. Once this connection is established, a database table is generated retrieving all the data from the database file. This ensures access to the database. Database views, namely the result sets of queries on the data stored in the database table, are also developed. Once manufacturing components, modelled at the micro-level agent, request access to a database view, a virtual table so-called database view table is generated. The data in this database view table are selected from the database table based on the requests made via the database view. The database view (table) concept is employed to categorise RFID data into different tables according to the location from which they are collected by the RFID system within the cryogenic supply chain. This facilitates a structured input of data into the multi-level agents of the model when requested (see Figure 3).

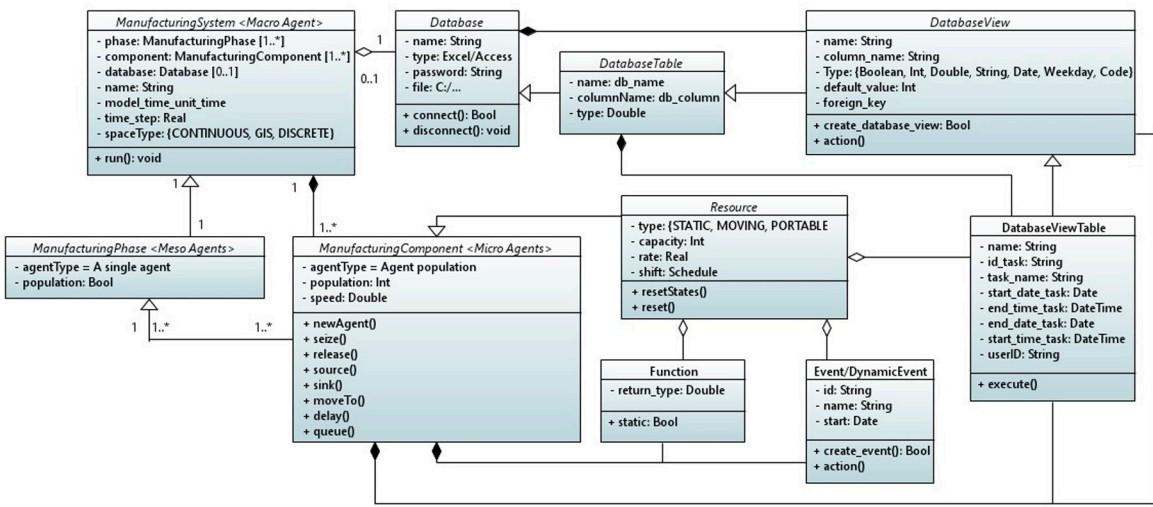

**Figure 3.** Unified Modelling Language (UML) class diagram of multi-agent cyber physical manufacturing system.

### 4.2. Cyber-Physical Architecture of IoT

An IoT data structure with three main layers of perception, transportation, and application is considered to demonstrate the CPMS architecture with RFID application. The three layers of CPMS-RFID system architecture are illustrated in Figure 4.

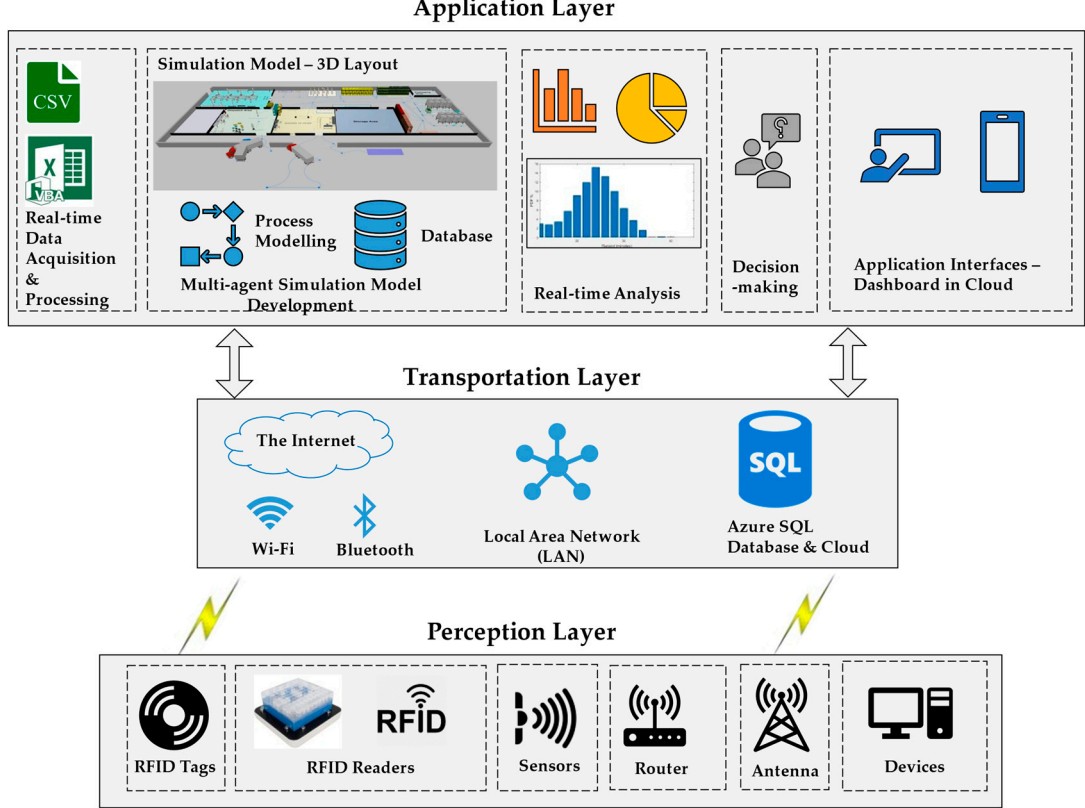

**Figure 4.** Cyber physical architecture of Internet of Things (IoT).

### 4.2.1. Perception Layer

The perception layer is employed for data collection via RFID tags. In this layer, various physical devices, including RFID tags and readers, sensors, router(s), antenna, cables, in-site computers and mobile devices such as laptops and tablets, are deployed. RFID tags are attached to the containers (e.g., vials and bags) that hold the sample media, as well as the packaging items (e.g., boxes used for the storage of vials, racks used for the storage of bags, and shippers) used for the transportation of the material within the cryogenic supply chain. Additionally, different types of RFID readers are designed and installed: (i) 'shipper readers' for scanning RFID tags attached to shippers; (ii) 'close proximity readers' for close up reads of bags and racks to automatically update the location of stored items without requiring manual data entry (typing), and (iii) 'cold box readers' for reading cryogenic vial boxes. In the latter RFID reader type, each vial stored in the box can be read individually as each vial slot provides a unique antenna.

The equipment used in the perception layer enables the collection of real-time information from the physical environment of the CPMS. The collected raw data include the complete track and trace, and the history of each stored sample from an assignment into storage, through audit checks and shipping events. The types of raw data gathered from the RFID readers situated in the cryogenic supply chain include task identification, supplier's and recipient's information or user name/ID once the RFID reading takes place within a site, date, time and location of RFID reading, order number, delivery location, carrier information, sample types, batch number, type of container, quantity and others. The perception layer playing a key role in the CPMS has a twofold purpose: firstly, to obtain data from the RFID sensors; and secondly, to execute operations by the command of the application layer. Raw data collected in the perception layer are sent to the network layer through the Internet.

### 4.2.2. Transportation Layer

The transportation layer is employed to connect all 'things', including anything like an object or a person for sharing and exchanging data. This layer supports the transferred real-time information through a wired or a wireless network from the perception layer to the application layer, and secondly the storage of the data captured from sensors. In this layer, data can be transmitted using local area network (LAN), the Internet, communications channels and private networks. In this research work, the Internet was selected due to its wide applicability, global availability and inexpensiveness. At each site within the cryogenic supply chain, there is a server PC that holds the system database (Microsoft SQL) for storing and processing the collected RFID data. Each server PC runs a web service that allows any device (laptop or tablet) connected to the same network to view and process the stored RFID records. The SQL database is hosted on a virtual machine on Microsoft Azure Cloud, enabling cross-site communication and providing easier interconnection between clients and manufacturers.

### 4.2.3. Application Layer

The application layer is the most interactive layer of the CPMS. It is employed to support a set of business services and realises intelligent computation and resources allocation in screening, selecting, producing and processing data. In this layer, the first step considers real-time data acquisition and processing, as seen in Figure 4. A Microsoft Excel file with real-time data is initially obtained. These data are collected from the RFID system as described in the perception layer and stored in the Azure SQL Database, as discussed in the transportation layer. This file is obtained via the Google Drive API and updated automatically at specific time intervals which can be configured by users via the Azure and Google Drive APIs. The data from this file are then processed automatically, by employing a set of macros with Excel Visual Basic for Applications (VBA), in order to create timestamps and calculate the time required for performing various tasks within the cryogenic supply chain. The next step focuses on multi-agent simulation model development that follows the multi-agent simulation technique discussed earlier in this work. Thus, a database to read real-time data obtained from the Excel file

and a simulation model that retrieves the data from the database and reflects the complex behaviour of manufacturing systems within a cryogenic supply chain are developed. Considering a cryogenic supply chain with the manufacturer, distributor and end-user, the input and output information flows to and from each location where RFID devices are implemented are seen in the UML composite structure diagram in Figure 5. The distributor acting as a cryogenic warehouse comprises three phases: Phase I—Receipt and Inventory, Phase II—Storage and Monitoring and Phase III—Distribution.

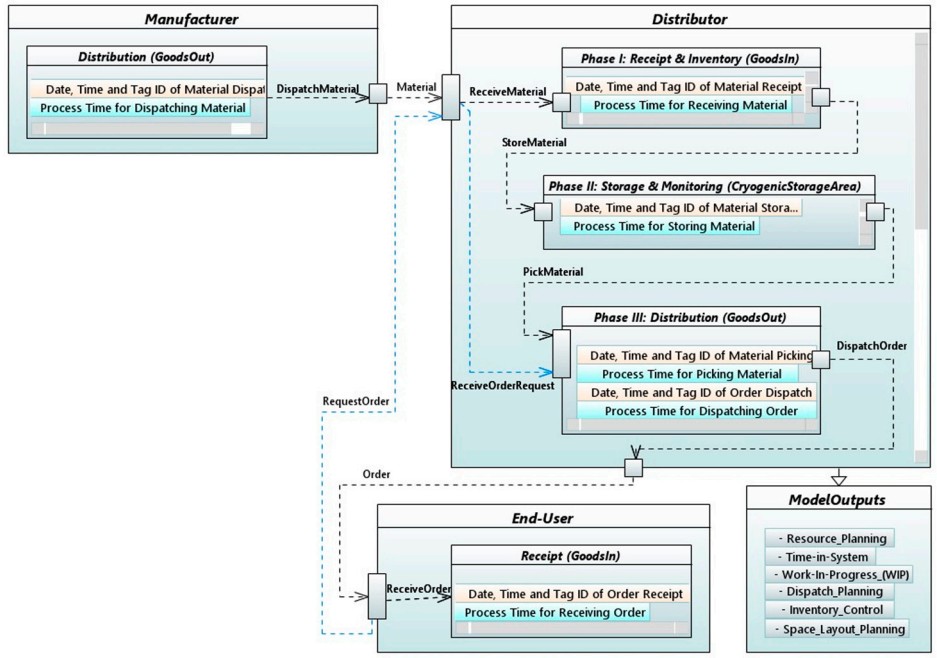

**Figure 5.** UML composite structure diagram of a cryogenic supply chain.

Manufacturer and end-user consider distribution and receipt tasks, respectively. The types of collected sensing data, obtained from the database to enable real-time simulation, include tags ID, date and time stamps recording after each tag is scanned by an RFID reader and process time required for each task to be carried out. After performing the model simulation, real-time data analysis and decision-making are introduced. In these steps of the application layer, the data obtained from either part of the simulation (each block in Figure 6) or the entire model are analysed. These data may include resource and space utilisation, lead times, time-in-system, system throughput, work-in-progress (WIP) and inventory size. Relevant data analytics can then be carried out to provide informed decisions in terms of resource planning, time-in-system, WIP, dispatch planning, inventory control and space layout planning. Finally, application interfaces—dashboard in *Cloud* transforms the developed simulation model into a decision-support platform allowing day-to-day operations. The developed simulation model can be stored in the Cloud and hence connected with operational data and set-up experiments providing decision-makers access to leverage simulation insights. Additionally, multiple users at different locations can remotely access the model, run simulations and visualise the results through interactive dashboards. Users are also able to perform various experiments and compare the results obtained in order to achieve better scenario management. A great benefit of the Cloud solution in the proposed cyber-physical architecture is the high-performance computing capabilities that they can provide by allowing the execution of complex multi-run experiments.

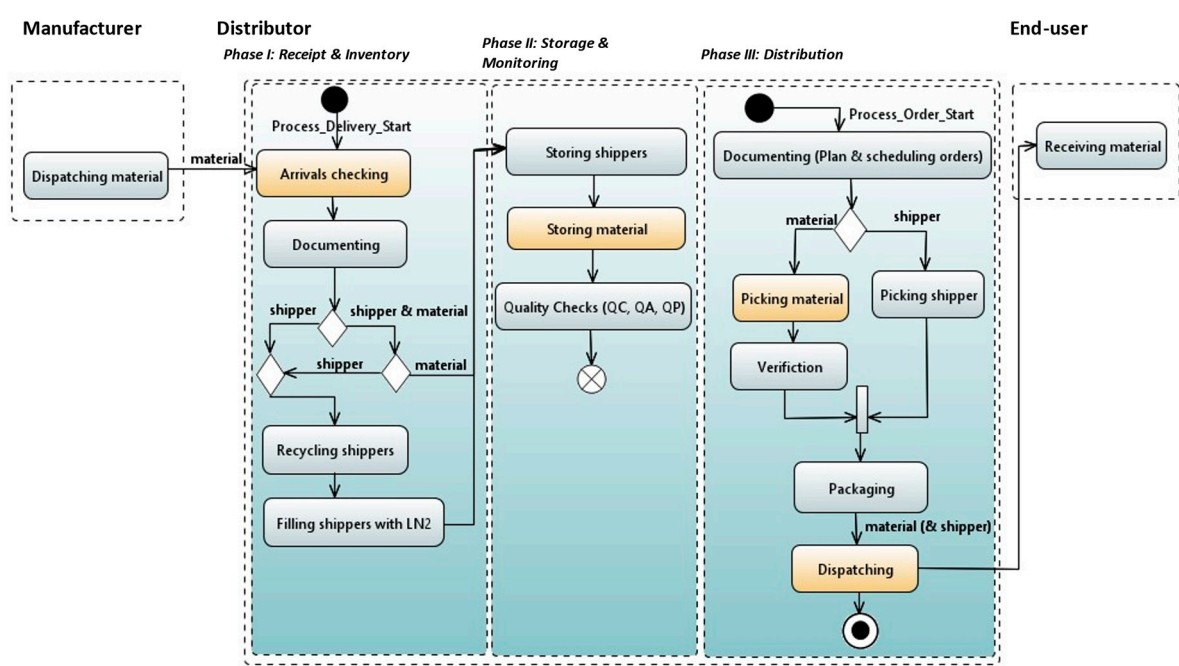

**Figure 6.** Case study: UML activity diagram for a cryogenic supply chain with RFID.

### 4.3. Result: A Case Study in Cryogenic Supply Chain

To demonstrate the applicability and validity of the proposed CPS-RFID architecture, a manufacturing system at a cell and gene therapy (CGT) cryogenic warehouse is studied. The manufacturing system is characterised by labour-intensive processes and repetitive tasks. The multiple parallel executions of interdependent tasks within different manufacturing phases increase the complexity of these systems. A dynamic simulation model in AnyLogic software (version 8) is developed to implement the presented CPS-RFID solution employing the agent-based technique. Simulating such a complex system following the proposed technique can simplify the complicated structure of the model. Additionally, RFID implementation can improve the efficiency and productivity of manufacturing systems in terms of system throughput, time-in-system and resource utilisation. The proposed CPS-RFID solution was applied to model the case study in order to identify the benefits of the RFID implementation within the selected manufacturing system.

#### 4.3.1. Outline of the System

The manufacturing system at the studied CGT cryogenic supply chain consists of three sites, a manufacturer, a distributor (cryogenic warehouse) and an end-user (patient). In this case study, the focus is on distributor role, who is responsible for receiving cryogenic material from the manufacturer, storing and monitoring the material, and dispatching it when requested from the end-user. The flow of material and information in the studied warehouse are modelled within three manufacturing phases of the system: (i) Receipt and Inventory; (ii) Storage and Monitoring; and (iii) Distribution. The tasks' execution within the cryogenic supply chain is explained with the help of the UML activity diagram shown in Figure 6.

RFID devices are implemented into the cryogenic warehouse, capturing information from three main warehouse areas: goods in, cryogenic storage area and goods out. Data produced by the RFID devices located in these areas are obtained from tasks including 'arrivals checking', 'storing material', 'picking material' and 'dispatching, as highlighted in Figure 6. Considering the implementation of the RFID system, the UML use case diagram of the studied cryogenic supply chain shown in Figure 7 represents the interactions of users in the cryogenic warehouse with manufacturers and end-users.

The exchange of information between the three sites, i.e., manufacturer, distributor and end-user, is carried out with the help of a unique identifier, an Autonomous System Number (ASN).

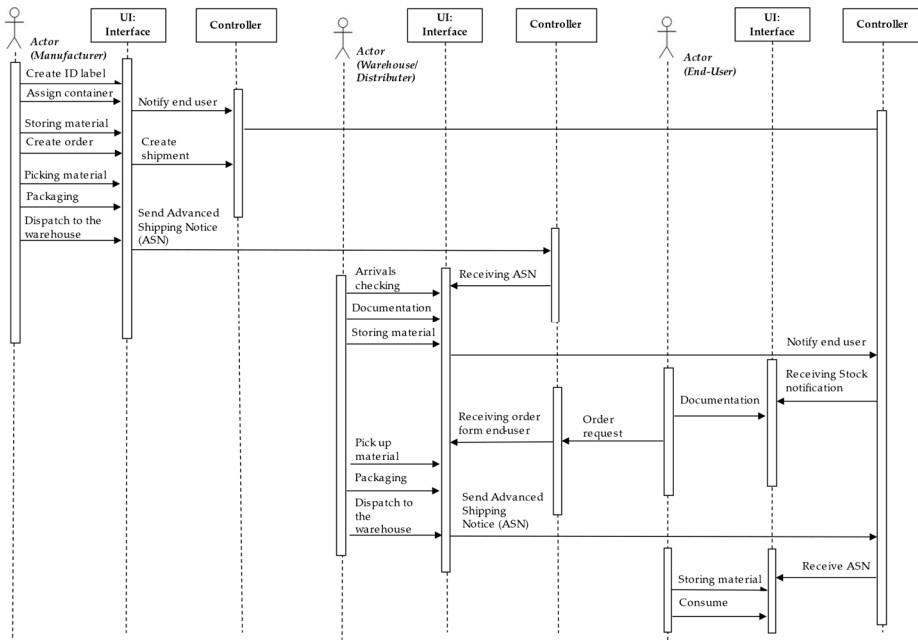

**Figure 7.** Case study: UML use case of cell and gene therapy (CGT) cryogenic supply chain.

### 4.3.2. Numerical Analysis

To develop the simulation model, data were collected for the CGT cryogenic storage processes of the cryogenic warehouse in the UK. Following the ABM-DES simulation method, the micro-level agents and the corresponding data are summarised in Table 1.

**Table 1.** Input data—micro-level agents.

| Agents | Technician | Recycle/Refill | QA | QC | QP | Cryocart | Trolley |
|---|---|---|---|---|---|---|---|
| Quantity | 16–20 | 4 | 4 | 6 | 2 | 4 | 2–3 |

Moreover, the cycle time distribution input for different processes is presented in Table 2. The tasks which will be affected by the RFID implementation are indicated in by an asterisk (*).

**Table 2.** Input data—cycle time distributions.

| Manufacturing Phase | Manufacturing Task | Distribution (min) |
|---|---|---|
| Phase I—Receipt and Inventory | Arrivals checking * | Uniform (2, 3) |
| | Documenting | Triangular (5, 7, 10) |
| | Recycling and Refilling shipper | Triangular (25, 47, 70) |
| | Storing/Picking material * | Triangular (10, 18, 30) |
| | Storing/Picking shipper * | Uniform (5, 10) |
| Phase II—Storage and Monitoring | QA quality check | Triangular (25, 30, 35) |
| | QC quality check | Triangular (15, 20, 25) |
| | QP quality check | Off-site, >1 day |
| | Documenting and Verification | Uniform (35, 55) |
| Phase III—Distribution | Packaging | Triangular (10, 15, 20) |
| | Dispatching * | Uniform (0.08, 0.17) |

4.3.3. Results: Model Validation

Validation of the proposed technique is accomplished using the data from the case study. Real data on the RFID cycle times for a five-week period were collected from the shop floor of the case study company. Average cycle times taken for each test procedure carried out within a trial are summarised in Table 3. According to these data, the continuous probability distribution for each RFID task is also found.

**Table 3.** RFID input data: Cycle times.

| (RFID) Task | Test Procedures | | | | | | | | Distribution |
|---|---|---|---|---|---|---|---|---|---|
| | 1 | 2 | 3 | 4 | 5 | 6 | 7 | 8 | |
| Arrivals Checking (s) | 10 | 10 | 10 | 15 | 15 | 15 | 15 | 15 | Uniform(10, 15) |
| Storing Material (min) | 2 | 2 | 2 | 4 | 4 | 4 | 4 | 4 | Uniform(2, 4) |
| Picking Material (min) | 1 | 1 | 3 | 3 | 3 | 1 | 3 | 1 | Uniform(1, 3) |
| Dispatching (s) | 10 | 10 | 15 | 15 | 15 | 10 | 15 | 10 | Uniform(10, 15) |

The cycle time distributions are implemented in the simulation model, and the average time-in-system for the three manufacturing phases for the 'RFID state' is obtained. The results are compared with the corresponding computational data obtained for the state 'Without RFID', and also the reductions in the average time-in-system are seen in Table 4.

**Table 4.** Average time-in-system for the states 'Without RFID' and 'With RFID'.

| Time-in-System for Phases I—III | 'Without RFID' State (min) | 'With RFID' State (min) | Reduction |
|---|---|---|---|
| Receipt and Inventory | 13 | 8 | 38.46% |
| Storage and Monitoring | 37 | 18 | 51.35% |
| Distribution | 79 | 33 | 58.23% |

From the computational results, it is seen that the RFID implementation can offer a significant reduction in the time required for storing and picking cryogenic material, as the system acts as an electronic witness. This eliminates the need for an extra person to act as a 'second witness'. Additionally, the system automates current processes by reducing the paperwork and documentation before dispatching the cryogenic material. Thus, RFID implementation reduces the time taken in filling out and making copies of forms, making resources simpler and safer. According to the simulation results, the efficiency and productivity of the cryogenic warehouse can be improved. The reduction observed in the cycle times (i) decreases the time-in-system, as shown in Table 4; (ii) can reduce the resource utilisation as the human and equipment resources are available for more time compared to the state without RFID; and (iii) can increase the number of deliveries from the manufacturer and orders from the end-user as operators are able to receive, store and dispatch a greater amount of materials.

## 5. Discussion

CPS modelling and design are being studied in the literature using different techniques and approaches such as mathematical algorithms [9], ontology-based approaches [23], DES methods [9] and the agent-based technique [5,12,25]. However, the existing literature has suggested that hybrid system design and engineering approaches are a suitable approach for the specification and analysis of CPS models due to the limitation of individual methods [45]. In the context of complex manufacturing and supply chain systems, mathematical modelling becomes complicated and computationally expensive due to the complexity and the high level of interactivity of sub-systems [11]. To tackle this challenge, the DES method provides a multi-agent simulation approach for CPS design and is a popular approach to simulate manufacturing processes. However, it limits the CPS architecture to a set of individual

layers and functions rather than taking the heterogeneous nature and interactivity of CPS architecture. In this regard, ontology-based and agent-based approaches provide the flexibility to create a multi-layer system engineering architecture for CPS design. Such bottom-up approaches require detailed and comprehensive information about complex systems and CPS architecture. The ABM technique takes an object-oriented programming approach to create a multi-layer system engineering architecture with multi agents for CPS design. This study proposed a hybrid multi-agent CPS architecture for complex manufacturing and supply chain systems. Such systems consist of several manufacturing phases and components that operate simultaneously and interact with each other over time. The concurrent processes and procedures can be labour-intensive with various types of manual tasks and quality checks. Therefore, the system performance is affected by human error and near misses significantly. To minimise such impact, a multi-agent cyber-physical system architecture with RFID technology is presented to assist inter-layer interactions between different manufacturing phases and among different stakeholders within a supply chain. In this context, the application of the multi-agent approach for CPS design has been previously studied for individual products [12,25], individual RFID functions [26], RFID control systems [5,33] and a set of products and machines with embedded RFID tags at a shop floor [29]. In this study, the proposed UML class diagram in Figure 3 presents the multi-agent CPS-RFID architecture. An agent-based simulation technique in the AnyLogic software was implemented to model the architecture. Moreover, the cyber-physical architecture of the IoT infrastructure is presented in Figure 4. A case study from a UK cryogenic supply chain in the CGT sector was selected to test the architecture and evaluate the impact of CPS-RFID on the supply chain performance. A number of UML diagrams are illustrated to present the developed multi-agent CPS-RFID architecture for the case study (see Figures 5–7). The performance is analysed based on the time-in-system for different manufacturing phases, as presented in Table 4. It is found that integrating RFID tags with appropriate IT platform is an effective approach in cryogenic warehouse management and can enable users to write and read information to and from the tags more effectively, during the cryogenic storage procedures. The outcome from this research shows that such capability has a clear impact on the performance of the supply chain by minimising the time-in-system by ~49.4%.

## 6. Conclusions

This paper extends the application of the agent-based approach and proposes a multi-agent CPS architecture for complex manufacturing and supply chain systems with embedded RFID technology. The proposed architecture is composed of a multi-layer agent at three different layers, called a macro-level agent for global manufacturing supply chain systems, meso-level for manufacturing phases, and micro-level for manufacturing components. RFID-based CPMS has been implemented at multiple levels within the global supply chain. A multi-layer data structure provides an appropriate IoT platform for the CPS-RFID system. A case study from the cryogenic supply chain has been selected to test the validity of the proposed architecture. Several UML diagrams are developed to present the different layers in the cyber-physical architecture of IoT. The proposed RFID-CPMS architecture shows that the multi-agent simulation model using the hybrid ABM-DES approach, together with the cyber-physical architecture of IoT infrastructure, is an effective approach to deploy a multi-agent CPS-RFID system architecture.

The further work will be focused on identifying different uncertainties within the multi-agent CPS-RFID architectures and developing an integrated DT-based CPS within the complex manufacturing and supply chain systems. Currently, the authors are working on studying the impact of RFID-based CPMS on resource utilisation and throughput. Further research can also be conducted to quantify this impact in the context of sustainability to evaluate the cost of goods and energy consumption.

**Author Contributions:** Conceptualization, M.F. and C.L.; methodology, M.F. and C.L.; software, M.F.; validation, M.F. and C.L.; formal analysis, M.F. and C.L.; investigation, M.F., C.L. and J.A.E.; resources, M.F., C.L. and J.A.E.; data curation, M.F., C.L.; writing—original draft preparation, M.F., C.L.; writing—review and editing, M.F., J.A.E.

and G.M.; visualisation, M.F., C.L. and G.M.; supervision, M.F. and J.A.E.; project administration, M.F. and J.A.E.; funding acquisition, J.A.E. All authors have read and agreed to the published version of the manuscript.

**Funding:** This research is funded by the Innovate UK [Grant No. 104515].

**Acknowledgments:** The authors would like to acknowledge the support form Cryogatt Systems Limited, ThermoFisher UK and Cell and Gene Therapy Catapult UK.

**Conflicts of Interest:** The authors declare no conflict of interest.

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
