# Peer review of "RFID Application in a Multi-Agent Cyber Physical Manufacturing System"

_jmmp, doi:10.3390/jmmp4040103_

Round 1

Reviewer 1 Report

Generally, a nice paper with a promising outcome.

There are however some weaknesses that need to be addressed:

The author limit the literature search, which is otherwise weil-organised, to cyber-physical systems and very similar variants. This does not nake to much sense as many research works concern CPS systems without ever using this notion. The authors must rethink the keywords and need to include also RFID production and logistics systems which are not refered to as CPS.

Furthermore, the authors state: "The proposed multi-agent RFID-CPS architecture is presented based on the literature review
and the theories behind RFID and CPS, as presented in Section 2". So the authors present a background and a solution - but not the rationale and processes for the synthesis in between - they should not just describe what they propose, but why they propose it. They need to explain, which alternative solution possibilities exist, which advantages the chosen solution possibilities have and in which systematic and documented process they were chosen. This concerns the whole sub-sections 4.1 and 4.2 - they need to be rewritten and the underlying causes for this kind of synthesis need to be explained.

Author Response

Please find below the response to the points raised, and please also find the attached revised manuscript.

Responses to Reviewers’ comments for the manuscript titled:

RFID Application in a Multi-agent Cyber Physical Manufacturing System

ID: jmmp-970371

The authors would like to thank the reviewers for their time, and their feedback that have provided us with the opportunity to improve the manuscript. We have endeavoured to break down our responses for each point highlighted by the Reviewers. The amendments are also highlighted with Track Changes in ‘red’ colour in the manuscript. We would also like to highlight that the manuscript has gone through another round of proof-reading, to eliminate any remaining typos or errors.

Review – 1

Generally, a nice paper with a promising outcome. There are however some weaknesses that need to be addressed:

Comment -1

The author limit the literature search, which is otherwise well-organised, to cyber-physical systems and very similar variants. This does not take to much sense as many research works concern CPS systems without ever using this notion. The authors must rethink the keywords and need to include also RFID production and logistics systems which are not referred to as CPS.

Authors’ Response:

Thank you for your comment. The authors would like to clarify that, this research work aims to address a specific research question: “How ABM technique can be applied to develop an RFID-CPMS architecture?” as highlighted in Section 1. Accordingly, a systematic review approach has been implemented to narrow down the relevant literature as detailed in Section 3. Thereby, the keyword CPS is not the only keyword which has been taken into account, but also any phrase begins with ‘cyber physical’, and ‘production system’ or ‘CPPS’ or ‘cyber physical production system’ and ‘RFID’ have also been considered. Therefore, the ‘theory’ section provided the background on an appropriate amount of literature to support the research gap and the contribution of the paper as detailed in Section 2. The relevant literature with the scope of production and logistics systems were already included e.g. Reference [1], [2], [4], [5], [6], [12], [15], [17], [26], [25], [27], [29], [34].

Comment -2

the authors state: "The proposed multi-agent RFID-CPS architecture is presented based on the literature review and the theories behind RFID and CPS, as presented in Section 2". So the authors present a background and a solution - but not the rationale and processes for the synthesis in between - they should not just describe what they propose, but why they propose it. They need to explain, which alternative solution possibilities exist, which advantages the chosen solution possibilities have and in which systematic and documented process they were chosen. This concerns the whole sub-sections 4.1 and 4.2 - they need to be rewritten and the underlying causes for this kind of synthesis need to be explained.

Authors’ Response:

Thank you for your comment. The authors would also like to clarify that. the rationale behind this research work has been presented in the ‘Introduction’ section and the research question: “How ABM technique can be applied to develop an RFID-CPMS architecture?” has been therefore defined. Moreover, further literature review to support the theory behind the research question with a view to highlight the contribution of this research work has been presented in Section 2. The answer to the question of ‘why’? is detailed in Section 1. paragraph 4 and 5 as in: “Acquiring essential data and maintaining information integrity within manufacturing processes and supply chains are challenging and time-consuming, due to the amount of paper works and manual audit procedures involved. In many small and medium enterprise (SMEs), the application of automated techniques such as IoT, CPS and RFID is not mature and is at early stages of proof-of-concept. Although such advancements create new challenges for industries in terms of uncertainty management [7], security [8], cloud computing [9] and big data management [10], evaluating the impact of such technologies on supply chains and businesses performance is crucial for any investment appraisal.

For a system of systems such as supply chains and complex manufacturing systems, the Agent-Based Modelling (ABM) technique has been a promising approach for the system design and system simulation [11]. To explore and substantiate the RFID application in CPMS, this paper addresses the following research question: “How ABM technique can be applied to develop an RFID-CPMS architecture?”. Moreover, this study contributes to knowledge by extending the application of ABM technique to develop a multi-agent architecture for an RFID-CPS. The context of this research is the application of RFID technology in a CPMS. We have also constructed a number of Unified Modelling Language (UML) diagrams to depict different perspectives and layers of the multi-agent RFID-CPS in a cryogenic supply chain.”

Moreover, the application and capability of ABM technique and hybrid simulation techniques to design and analyse supply chains and complex systems have been studied by many authors such as Julka et al. 2002; Kaihara 2003; Jiao et al 2009; Loau et al 2011; Mathiew et al 2018; Farsi et al 2019; and many others. The authors would like to clarify that the main contribution of this manuscript is to extend the application of ABM technique to develop a multi-agent architecture for an RFID-CPS. The authors would also like to clarify that this manuscript would not imply that the proposed hybrid approach is the only approach in this field of study. As suggested, and for further clarification, Discussion in Section 5, Paragraph 1. has been extended to justify the proposed approach and include the limitation and advantages of the proposed technique, as in:  

CPS modelling and design are being studied in the literature using different techniques and approaches such as mathematical algorithm [9], ontology-based approaches [23], DES methods [9] and agent-based technique [5], [12], [25]. However, the existing literature has suggested that hybrid system design and engineering approaches are a suitable approach for the specification and analysis of CPS models due to the limitation of individual methods [36]. In the context of complex manufacturing and supply chain systems, mathematical modelling becomes complicated and computationally expensive due to the complexity and the high level of interactivity of sub-systems [11]. To tackle this challenge, the DES method provides a multi-agent simulation approach for CPS design and is a popular approach to simulate manufacturing processes. However, it limits the CPS architecture to a set of individual layers and functions rather than taking the heterogeneous nature and interactivity of CPS architecture. In this regard, ontology-based and agent-based approaches provide the flexibility to create a multi-layer system engineering architecture for CPS design. Such bottom-up approaches require detailed and comprehensive information about complex systems and CPS architecture. ABM technique takes an object-oriented programming approach to create a multi-layer system engineering architecture with multi agents for CPS design.”

Reviewer 2 Report

This document proposes a multi-agent CPS architecture for complex manufacturing and supply chain systems with embedded RFID technology. The analysis of the results of the proposed architecture simulation shows an improvement in efficiency and productivity, in terms of time resources in the system.
The topic of the work may be of interest in facilitating interactions between different manufacturing stages on the shop floor and external interactions with other stakeholders within a supply chain.

I would like to make the following suggestions:
- I would include "Cell and Gene Therapy" in the Keywords.
- On line 186 the unit "ËšC" is not in the correct format.
- It is possible to indicate the influence that the system would have on other aspects other than time: such as the number of errors avoided, impact on workers, economic investment, energy consumption...
- I recommend placing the DOI to the references used.
- The information in reference 36 is incomplete. What is it?

Author Response

Please find below the response to the points raised, and please also find the attached revised manuscript.

Responses to Reviewers’ comments for the manuscript titled:

RFID Application in a Multi-agent Cyber Physical Manufacturing System

ID: jmmp-970371

The authors would like to thank the reviewers for their time, and their feedback that have provided us with the opportunity to improve the manuscript. We have endeavoured to break down our responses for each point highlighted by the Reviewers. The amendments are also highlighted with Track Changes in ‘red’ colour in the manuscript. We would also like to highlight that the manuscript has gone through another round of proof-reading, to eliminate any remaining typos or errors.

Review – 2

This document proposes a multi-agent CPS architecture for complex manufacturing and supply chain systems with embedded RFID technology. The analysis of the results of the proposed architecture simulation shows an improvement in efficiency and productivity in terms of time resources in the system. The topic of the work may be of interest in facilitating interactions between different manufacturing stages on the shop floor and external interactions with other stakeholders within a supply chain. I would like to make the following suggestions:

Comment -1

- I would include "Cell and Gene Therapy" in the Keywords.

Authors’ Response:

Thank you for your comment. The keyword has been added as suggested.

Comment -2

- On line 186 the unit "ËšC" is not in the correct format.

Authors’ Response:

Thank you for your comment. The unit format has been amended as suggested.

Comment -3

- It is possible to indicate the influence that the system would have on other aspects other than time: such as the number of errors avoided, impact on workers, economic investment, energy consumption...

Authors’ Response:

Thank you for your comment. It is sensibly remarked by the reviewer. Further work has been expanded to highlight these aspects. Section 6. Last paragraph, “… Currently, the authors are working on studying the impact of RFID-based CPMS on resource utilisation, inventory management and throughput. Further research can also be conducted to quantify this impact in the context of sustainability to evaluate the cost of goods, waste and energy consumptions.

Comment -4

- I recommend placing the DOI to the references used.

Authors’ Response:

Thank you for your comment. The available DOIs have been added to the reference.

Comment -5

- The information in reference 36 is incomplete. What is it?

Authors’ Response:

Thank you for your comment. This is the reference to an internal technical report from the sponsor Cryogatt company who provided the case study as part of the Innovate UK funded project. (please see the acknowledgement). However, since the lead member of the company is one of the co-authors in this manuscript, the authors decided to remove this reference.

Round 2

Reviewer 1 Report

The addtion of the new section clarifies the contribution and allows, form my point of view, the publication.